 eLife

# Diverse nucleosome Site-Selectivity among histone deacetylase complexes

**Zhipeng A Wang[1,2†], Christopher J Millard[3†], Chia-Liang Lin[3†], Jennifer E Gurnett[3], Mingxuan Wu[1,2], Kwangwoon Lee[1,2], Louise Fairall[3], John WR Schwabe[3]\*, Philip A Cole[1,2]\***

[1]Division of Genetics, Department of Medicine, Brigham and Women's Hospital, Boston, United States; [2]Department of Biological Chemistry and Molecular Pharmacology, Harvard Medical School, Boston, United States; [3]Leicester Institute of Structural and Chemical Biology, Department of Molecular and Cell Biology, University of Leicester, Leicester, United Kingdom

**Abstract** Histone acetylation regulates chromatin structure and gene expression and is removed by histone deacetylases (HDACs). HDACs are commonly found in various protein complexes to confer distinct cellular functions, but how the multi-subunit complexes influence deacetylase activities and site-selectivities in chromatin is poorly understood. Previously we reported the results of studies on the HDAC1 containing CoREST complex and acetylated nucleosome substrates which revealed a notable preference for deacetylation of histone H3 acetyl-Lys9 vs. acetyl-Lys14 (Wu et al, 2018). Here we analyze the enzymatic properties of five class I HDAC complexes: CoREST, NuRD, Sin3B, MiDAC and SMRT with site-specific acetylated nucleosome substrates. Our results demonstrate that these HDAC complexes show a wide variety of deacetylase rates in a site-selective manner. A Gly13 in the histone H3 tail is responsible for a sharp reduction in deacetylase activity of the CoREST complex for H3K14ac. These studies provide a framework for connecting enzymatic and biological functions of specific HDAC complexes.

**\*For correspondence:**
john.schwabe@le.ac.uk (JWRS);
pacole@bwh.harvard.edu (PAC)

[†]These authors contributed equally to this work

## Introduction

Reversible histone acetylation (*Cole, 2008*) regulates a range of major fundamental biological processes including gene expression, DNA replication, cell growth, and differentiation (*Jamaladdin et al., 2014*). Histone acetyltransferases (HATs) catalyze Lys sidechain acetylation on the N-terminal tails of each of the four core histones (*Weinert et al., 2018*), H2A, H2B, H3, and H4 that comprise nucleosomes, the building blocks of chromatin (*Helin and Dhanak, 2013*; *Figure 1A*). Enzymological and cellular analysis has shown that particular HATs such as p300/CBP, PCAF/GCN5, and Myst family members display varying degrees of site selectivity in the context of histone tail peptides and/or chromatin (*Shortt et al., 2017*). For example, p300/CBP is best known to acetylate Lys18 and Lys27 in histone H3 (*Dancy and Cole, 2015*), whereas PCAF/GCN5 prefers to acetylate Lys14 and Lys9 of histone H3 (*Ali et al., 2018*). Histone deacetylases (HDACs) remove acetyl-Lys modifications in chromatin and other proteins and belong to two major families in humans (*Taunton et al., 1996*), classical HDACs which include 11 Zn metallohydrolase enzymes and seven sirtuins (*Wang et al., 2017*), that catalyze NAD-dependent Lys deacetylation (*Falkenberg and Johnstone, 2014*). The class I HDACs, HDAC1, HDAC2, and HDAC3 are relatively similar at the amino acid sequence level and show overlapping pharmacological sensitivities to the benzamide class of HDAC inhibitors represented by entinostat (*Yang et al., 2019*).

HDAC1 and HDAC2 (*Kelly and Cowley, 2013*) are the most similar HDACs and are interchangeably found in a variety of multisubunit protein complexes including CoREST, Sin3(A/B), NuRD, and MiDAC whereas HDAC3 is commonly associated with the SMRT/NCoR complex (*Bantscheff et al.,*

*2011*; *Figure 1B*). The CoREST (*Humphrey et al., 2001*), Sin3 (*Laherty et al., 1997*), NuRD (*Xue et al., 1998*), (*Zhang et al., 1999*), and SMRT (*Guenther et al., 2000*), (*Wen et al., 2000*), (*Li et al., 2000*) complexes are generally believed to be involved in silencing of gene expression and are recruited by various transcription factors to chromatin in a cell type and cell state dependent manner (*Ooi and Wood, 2007*). Less is known about the MiDAC complex although it has been associated with mitotic processes (*Bantscheff et al., 2011*). Recent structural studies on each of these five HDAC complexes have provided insights into their distinct molecular architectures (*Millard et al., 2017*) (CoREST (*Song et al., 2020*), NuRD (*Millard et al., 2016*), Sin3A (*Clark et al., 2015*), MiDAC (*Itoh et al., 2015*), and SMRT (*Watson et al., 2012a*) and inositol phosphates have been shown to have varying roles in regulating their stabilities and catalytic activities (*Millard et al., 2013*).

Histone substrate site-selectivity experiments with HDAC enzymes have generally involved analyzing the isolated HDAC polypeptides or catalytic domains and acetyl-Lys containing histone peptide tails rather than the more physiological HDAC complexes and nucleosome substrates (*Zhang et al.,*

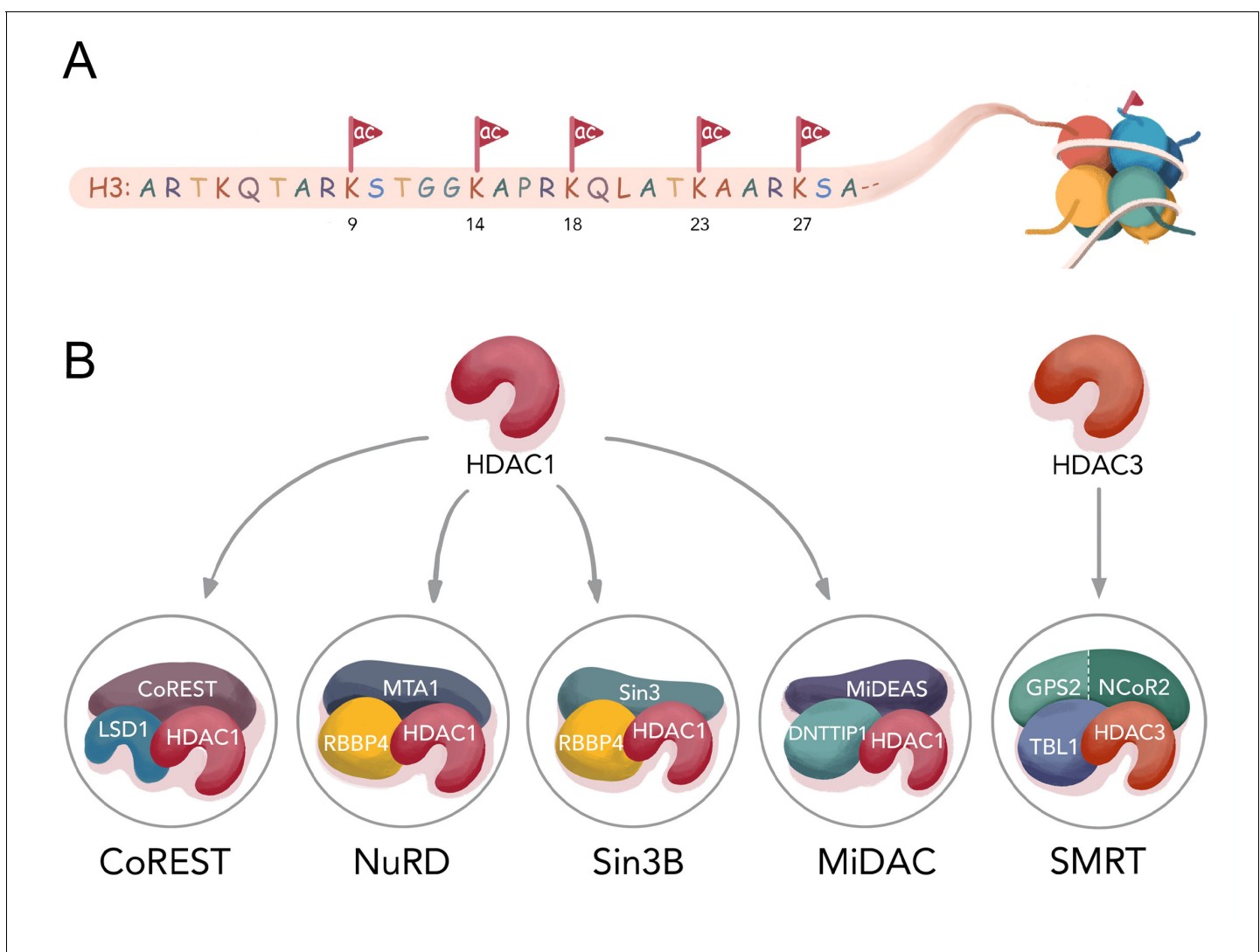

**Figure 1.** Histone H3 tail acetylation and HDAC complexes. (**A**) Different acetylation sites on H3 N-terminal tail studied in this manuscript; (**B**) Components of four well- established HDAC1 complexes CoREST (LSD1, HDAC1, CoREST1), NuRD (MTA1, HDAC1, RBBP4), Sin3B (Sin3, HDAC1, RBBP4), MiDAC (MIDEAS, HDAC1, DNTTP1), and one HDAC3 complex SMRT (GPS2-NCoR2 chimera, HDAC3, and TBL1 ).

2016), (*Riester et al., 2007*), (*Gurard-Levin et al., 2009*). Such studies have typically shown little to no amino acid sequence site-selectivities for HDACs. As the physiological enzyme forms of HDAC1-3 exist in multisubunit complexes (*Millard et al., 2017*) and the natural substrates are chromatin comprised of nucleosomes, histone octamers wrapped by 146 bp of double stranded DNA, the biological relevance of the free HDAC and histone tail substrate peptide experiments is unclear. Recent studies on the CoREST complex and acetylated nucleosome substrates revealed a notable preference for deacetylation of histone H3 acetyl-Lys9 vs. acetyl-Lys14 (*Wu et al., 2018*). This was especially interesting because the CoREST complex also includes the enzyme LSD1 which specifically demethylates methyl-Lys4 in histone H3 and this activity is strongly inhibited when the H3 tail also contains acetyl-Lys14 but not acetyl-Lys9 (*Kalin et al., 2018*). These properties suggest a biological role in histone H3 marked by Lys14 acetylation and Lys4 methylation to protect against CoREST complex silencing. The CoREST complex enzymatic studies also raise the possibility that different HDAC complexes may show distinct deacetylase kinetics and nucleosome site selectivities.

In this study, we evaluate the catalytic actions of five purified class I HDAC complexes CoREST, MiDAC, NuRD, Sin3B, and SMRT with five different histone H3 acetylated mononucleosomes as well as the corresponding free histone H3 substrates. We find that the kinetics of the HDAC complexes across the range of free acetylated histone H3 substrates are fairly similar, whereas there are large differences between the complexes with respect to deacetylase rates and site-selectivity with nucleosome substrates. Among these complexes, the CoREST complex shows a special resistance to H3K14ac which we find is largely driven by the preceding Gly13 residue.

## Results

### Production of H3 acetylated nucleosomes and HDAC complexes

Semisynthetic *X. laevis* histone H3 proteins (*Wang et al., 2015*) mono-acetylated at positions Lys9, Lys14, Lys18, Lys23, and Lys27 were prepared using F40 sortase (*Piotukh et al., 2011*). In this approach, the N-terminal tails aa1-34 were prepared as synthetic peptides containing the acetyl-Lys and terminating in a depsipeptide linkage between Thr and Gly and the H3 globular domain aa34-135 was produced recombinantly. F40 sortase treatment of the H3 peptide and H3 globular domain catalyzes transpeptidation leading to ligation of the fragments to produce pure, scarless full-length modified histone H3s (*Figure 2A–D* and *Figure 2—figure supplements 1–2*). Western blot analysis with the site-specific relevant acetyl-Lys antibodies demonstrated that each of the semisynthetic histone H3s contained the designated marks (*Figure 3—figure supplement 1* and *Wu et al., 2018*). The semisynthetic acetylated H3s were incorporated into mononucleosomes containing 146 bp DNA 601 Widom sequence (*Luger et al., 1997*; *Figure 2—figure supplement 3A–B*). The HDAC complexes CoREST (LSD1, HDAC1, CoREST1), NuRD (MTA1, HDAC1, RBBP4), Sin3B (Sin3, HDAC1, RBBP4), MiDAC (MIDEAS, HDAC1, DNTTIP1), and SMRT (GPS2-NCOR2 chimera, HDAC3, and TBL1) were expressed in HEK293F cells by transient transfection of three plasmids encoding the relevant proteins (*Figure 3—figure supplement 2A*). The details of how these complexes have been arrived at and are produced have been described previously (*Song et al., 2020*), (*Millard et al., 2016*), (*Clark et al., 2015*), (*Itoh et al., 2015*), (*Watson et al., 2012a*), (*Zhang et al., 2018*), (*Watson et al., 2012b*), (*Portolano et al., 2014*). In general, the two non-HDAC proteins in each case were selected based on the following criteria: 1) A set of proteins that included both well-established HDAC and nucleosome binding partners for a given complex, 2) Efficient transient co-expression of soluble proteins in HEK293F that stay associated by immunoaffinity chromatography and lead to relatively pure and concentrated complexes in peak fractions (>50% purity), 3) The ability of the core complexes to be reproducibly isolated as monodisperse peaks in appropriate stoichiometries after size exclusion chromatography. The HDAC complexes employed here were shown to be relatively pure and in the expected stoichiometries by SDS-PAGE (*Figure 3—figure supplement 2B*).

### Deacetylase assays with HDAC complexes

Each of the HDAC complexes was assayed with isolated acetylated histone H3 proteins or acetylated nucleosomes as substrates, using Western blot to monitor the disappearance of the requisite acetyl-Lys modification. The five HDAC complexes displayed robust catalytic activities with isolated

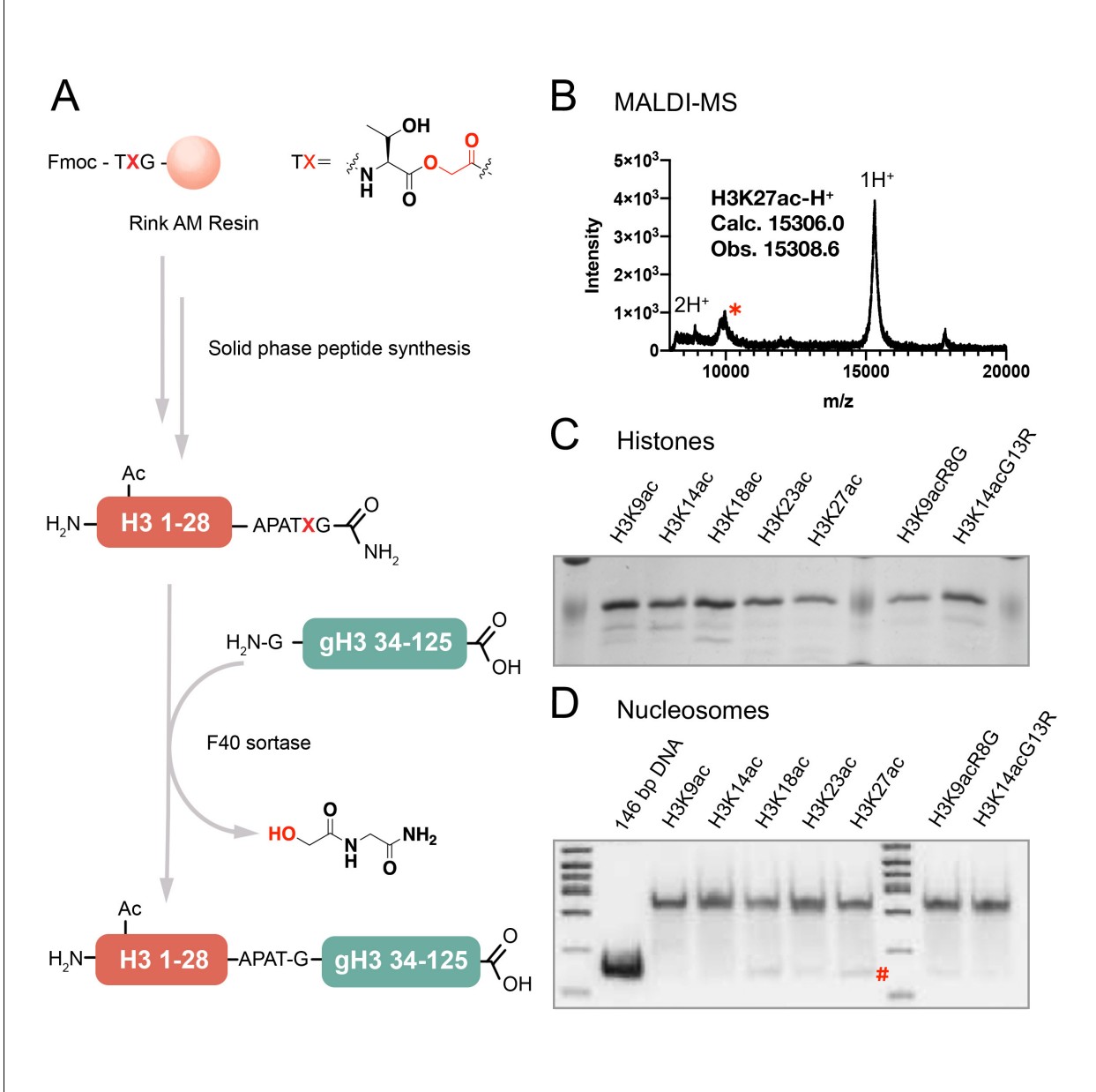

**Figure 2.** The semi-synthesis of full-length histone H3 with site-specific acetylations. (A) Synthesis of H3 proteins with site-specific acetylations; gH3: globular region of histone H3; (B) MALDI-MS for a semi-synthetic histone H3 product, using H3K27ac as an example, *: an unknown minor impurity; (C) SDS-PAGE of all the H3 proteins with acetylations, as H3K9ac, H3K14ac, H3K18ac, H3K23ac, H3K27ac, H3K9acR8G, H3K14acG13R; (D) Native 6% TBE gel of the nucleosome folding results with acetylated H3s, each showing ≥95% purity. #: minor free DNA band.

The online version of this article includes the following figure supplement(s) for figure 2:

**Figure supplement 1.** MALDI-TOF spectra for H3K9/14/18/23/27ac 1–34 TOG peptides.
**Figure supplement 2.** MALDI-TOF spectra for H3K9/14/18/23/27ac full length histone proteins.
**Figure supplement 3.** The assembly of H3 proteins with acetylations into corresponding nucleosomes.

acetylated H3 substrates with velocities (V/E) averaging in the range of 0.5–2 min$^{-1}$ and modest differences (largely <3 fold) in acetyl-Lys site-selectivity among K9ac, K14ac, K18ac, K23ac, and K27ac. These HDAC complexes show similar deacetylase activities as the isolated commercial HDAC1 (averaging ~0.8 min$^{-1}$) with free histone H3 substrates. These results are generally consistent with the prior studies that have indicated that local amino acid context within peptide substrates does

not have a large impact on HDAC-catalyzed deacetylation kinetics (*Table 1*, *Figure 4—figure supplements 1–6A–E,G*).

In contrast to the results with isolated H3 substrates, the results with acetylated nucleosome substrates showed dramatic differences among the HDAC complexes. The HDAC complex velocities (V/E) with acetylated nucleosomes varied over 200-fold, ranging from <0.005 min$^{-1}$ to 1.2 min$^{-1}$. The MiDAC complex showed the fastest velocities with V/E up to 1.2 min$^{-1}$ whereas the NuRD and Sin3B complexes showed V/E ~ 0.007 min$^{-1}$. The CoREST and SMRT complexes were intermediate with V/E values averaging in the range of 0.03–0.1 min$^{-1}$ (*Figure 3*).

Among the five HDAC complexes, there were distinct substrate selectivity features that are notable. CoREST showed similar velocities (V/E 0.06–0.1 min$^{-1}$) for K9ac, K18ac, K23ac, and K27ac that were markedly greater than the V/E for K14ac (<0.005 min$^{-1}$). In contrast, the MiDAC complex velocity for deacetylating K9ac (V/E 1.2 min$^{-1}$) was only about 2-fold greater than for K14ac (V/E 0.52 min$^{-1}$). However, the MiDAC complex deacetylated K23ac at a much lower rate (V/E 0.048 min$^{-1}$) relative to K9ac removal and in general, MiDAC deacetylase activity decreased when comparing more C-terminal acetyl-Lys sites in H3 with more N-terminal locations (*Figure 4A–F*). The one HDAC3 complex investigated here, SMRT, showed the highest rates of deacetylase activities on K9ac and K27ac and in general had more similar rates across the five sites compared with CoREST and MiDAC. The activities of NuRD and Sin3B displayed very low similar deacetylase activity at every H3 acetyl-Lys site (*Table 2*, *Figure 4—figure supplements 1–6A–F*).

## CoREST and MiDAC analysis on mutant acetylated nucleosomes

Three out of the five acetyl-Lys positions in the H3 tail examined here contain an Arg directly preceding the Lys (K9, K18, and K27) whereas Lys14 is preceded by Gly, the smallest amino acid. Given that among the five HDAC complexes studied here, K14ac cleavage from nucleosomes is in all cases slower than K9ac (most dramatically with the CoREST complex) we swapped amino acid residues at the 8 and 13 positions in H3. That is, we prepared K9ac nucleosomes containing R8G and K14ac nucleosomes bearing G13R (*Figure 5*, *Figure 5—figure supplements 1* and *2–3A–D*). Fortunately, the corresponding K9ac and K14ac selective antibodies were still able to recognize these mutant nucleosomes and could be used in HDAC assays. We performed deacetylase assays on these nucleosomes using the two most robust nucleosome HDAC complexes in our hands, the CoREST complex and MiDAC complex. The kinetic measurements with the CoREST complex showed striking differences relative to WT acetylated nucleosomes. The K9acR8G nucleosomes displayed a ~ 3 fold deacetylase rate reduction compared with WT K9ac (K9acR8G V/E 0.029 min$^{-1}$ vs WT K9ac V/E 0.1 min$^{-1}$). Remarkably, K14acG13R nucleosomes showed a > 17 fold increase in deacetylase rate vs. WT K14ac (K14acG13R V/E 0.084 min$^{-1}$ vs. WT K14ac V/E < 0.005 min$^{-1}$). The near matching of V/E deacetylase values of the CoREST complex on K14acG13R and WT K9ac nucleosomes suggests a profound rejection of a Gly at the H3 13th position in nucleosomes by the CoREST complex although it is more tolerated at the 8 position of H3. The kinetic studies with the MiDAC complex showed much

**Table 1.** Rate constants of different HDAC1/HDAC3 complexes on various histone proteins with site-specific acetylation.

Kinetic values shown are ± S.E.M. For each enzymatic reaction on histone proteins: the semi-synthetic H3K9/14/18/23/27ac histone H3 proteins (1.0 µM) were incubated with different HDAC1 and HDAC3 complexes at the following concentrations (2 nM CoREST, 10 nM MiDAC, 5 nM NuRD, 4 nM Sin3B, 2 nM SMRT, and 5 nM HDAC1, n = 2–5) under a reaction buffer of 50 mM HEPES 7.5 containing 100 mM KCl, 100 µM IP6, and 0.2 mg/mL BSA at 37 °C. The reaction time was counted, and multiple samples were taken at different time points between 0–30 min. The complexes include CoREST, NuRD, Sin3B, MiDAC, and SMRT, together with free HDAC1 enzyme. The kinetic data were divided by a factor of 5 in order to be comparable with the kinetic data on nucleosomes.

| Normalized V/[E] (min$^{-1}$) | HDAC1 | CoREST(RCOR1) | MiDAC(MiDEAS) | NuRD(MTA1) | Sin3B(Sin3) | SMRT(NCoR2) |
|---|---|---|---|---|---|---|
| H3K9ac | 0.30 ± 0.046 | 2.1 ± 0.38 | 0.42 ± 0.086 | 0.84 ± 0.38 | 0.56 ± 0.15 | 0.44 ± 0.10 |
| H3K14ac | 0.65 ± 0.40 | 2.8 ± 0.18 | 0.79 ± 0.29 | 1.5 ± 0.58 | 1.1 ± 0.093 | 0.78 ± 0.34 |
| H3K18ac | 1.1 ± 0.27 | 1.8 ± 0.37 | 0.45 ± 0.29 | 0.86 ± 0.32 | 1.2 ± 0.24 | 1.5 ± 0.57 |
| H3K23ac | 0.64 ± 0.10 | 2.6 ± 0.55 | 0.22 ± 0.037 | 0.36 ± 0.12 | 0.93 ± 0.20 | 0.65 ± 0.32 |
| H3K27ac | 1.1 ± 0.25 | 2.0 ± 0.35 | 0.43 ± 0.038 | 1.5 ± 0.48 | 1.1 ± 0.31 | 1.8 ± 0.80 |

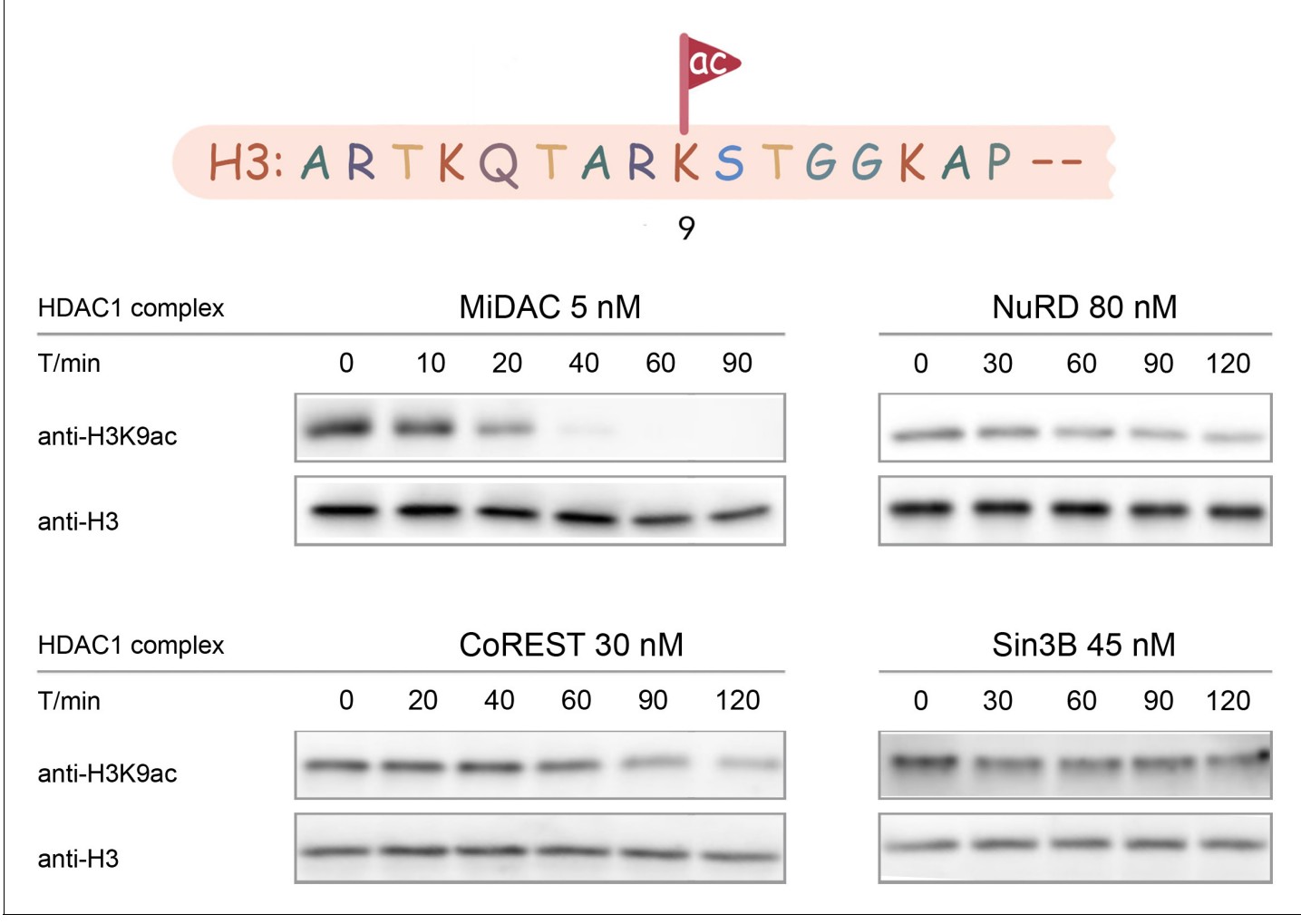

**Figure 3.** Typical results for Western-blot based deacetylation activity assay using different complexes on nucleosome with H3K9ac (n = 2 for each assay).

The online version of this article includes the following figure supplement(s) for figure 3:

**Figure supplement 1.** Primary antibody specificity test for anti-H3K23ac and anti-H3K27ac.
**Figure supplement 2.** The expression of HDAC complexes.

less sensitivity to changes in Arg8 and Gly13. In fact, K9acR8G nucleosome deacetylation by the MiDAC complex matched that of WT K9ac nucleosomes (V/E 1.2 min$^{-1}$ for both). K14acG13R nucleosome deacetylation by MiDAC was accelerated by 2.5 fold relative to WT K14ac nucleosomes, indicating a minor preference for Arg at the H3 13th position.

## NuRD (HDAC1-MTA1-RBBP4) deacetylation of H4K16ac nucleosomes

The very slow rates of nucleosome H3 deacetylation by the RBBP4 containing NuRD and Sin3B complexes prompted us to consider that the scaffold protein RBBP4 might be impeding the catalytic activity by obstructing the H3 tails (*Millard et al., 2016*). We thus examined whether a nucleosome containing acetylation on histone H4 might be an improved substrate for an RBBP4 complex. However, the deacetylation rate (V/E) for NuRD was low and similar to those of the acetylated H3 nucleosomes (*Figure 6*, *Figure 6—figure supplement 1A,B*). These results suggest that the low nucleosome deacetylase activity of RBBP4-containing HDAC complexes may be unrelated to RBBP4/H3 interaction.

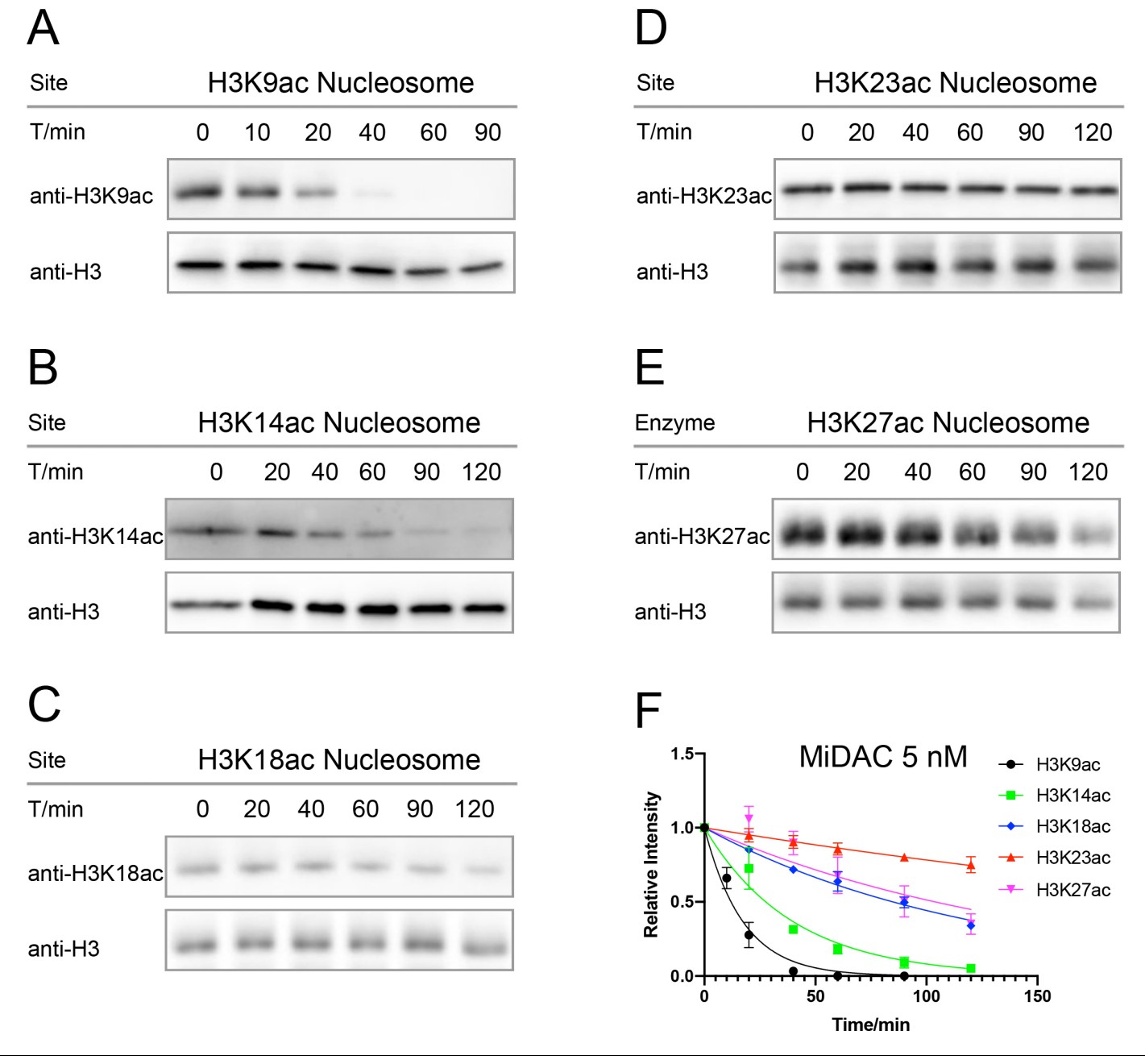

**Figure 4.** Typical results for Western-blot based deacetylation activity assay with 5 nM MiDAC on different nucleosomes with site-specific acetylation. (A) nucleosome with H3K9ac, n = 2, V/E = 1.2 ± 0.19 min$^{-1}$; (B) nucleosome with H3K14ac, n = 3, V/E = 0.52 ± 0.034 min$^{-1}$; (C) nucleosome with H3K18ac, n = 2, V/E = 0.17 ± 0.0051 min$^{-1}$; (D) nucleosome with H3K23ac, n = 4, V/E = 0.048 ± 0.0071 min$^{-1}$; (E) nucleosome with H3K27ac, n = 5, V/E = 0.18 ± 0.036 min$^{-1}$, and (F) the curve fitting for kinetics analysis.

The online version of this article includes the following figure supplement(s) for figure 4:

**Figure supplement 1.** CoREST (CoREST-LSD1-HDAC1) activity assay over nucleosomes or histones with acetylation at different sites.
**Figure supplement 2.** MiDAC (MiDEAS-DNTTIP1-HDAC1) activity assay over nucleosomes or histones with acetylation at different sites.
**Figure supplement 3.** NuRD (MTA1-RBBP4-HDAC1) activity assay over nucleosomes or histones with acetylation at different sites.
**Figure supplement 4.** Sin3B (Sin3-HDAC1-RBBP4) activity assay over nucleosomes or histones with acetylation at different sites.
**Figure supplement 5.** SMRT (GPS2-NCoR2 chimera-HDAC3-TBL1) activity assay over nucleosomes or histones with acetylation at different sites.
**Figure supplement 6.** HDAC1 activity assay over nucleosomes or histones with acetylation at different sites.

**Table 2.** Rate constants of different HDAC1 and HDAC3 complexes on various nucleosomes with site-specific acetylation on H3. Kinetic values shown are ± S.E.M. For each enzymatic reaction on nucleosomes: the semi-synthetic H3K9/14/18/23/27ac nucleosomes (100 nM) were incubated with different HDAC1 and HDAC3 complexes at the following concentrations (30 nM CoREST, 5 nM MiDAC, 80 nM NuRD, 45 nM Sin3B, 60 nM SMRT, and 60 nM HDAC1, n = 2–5), under a reaction buffer of 50 mM HEPES 7.5 containing 100 mM KCl, 100 μM IP6, and 0.2 mg/mL BSA at 37 ˚C. The reaction time was counted, and multiple samples were taken at different time points generally between 0–120 min (The MiDAC complex on H3K9ac was between 0–90 min). The complexes include CoREST, NuRD, Sin3B, MiDAC, and SMRT, together with free HDAC1 enzyme.

| V/[E] (min$^{-1}$) | HDAC1 | CoREST(RCOR1) | MiDAC(MiDEAS) | NuRD(MTA1) | Sin3B(Sin3) | SMRT(NCoR2) |
|---|---|---|---|---|---|---|
| H3K9ac | N.D. | 0.10 ± 0.012 | 1.2 ± 0.19 | 0.012 ± 0.0032 | 0.0050 ± 0.0043 | 0.046 ± 0.0045 |
| H3K14ac | N.D. | <0.005 | 0.52 ± 0.034 | 0.0091 ± 0.00080 | <0.005 | 0.019 ± 0.0022 |
| H3K18ac | N.D. | 0.062 ± 0.0019 | 0.17 ± 0.0051 | 0.0046 ± 0.00088 | 0.0082 ± 0.0030 | 0.021 ± 0.0039 |
| H3K23ac | N.D. | 0.067 ± 0.010 | 0.048 ± 0.0071 | 0.0071 ± 0.00083 | 0.0084 ± 0.0033 | 0.0081 ± 0.0028 |
| H3K27ac | N.D. | 0.093 ± 0.0088 | 0.18 ± 0.036 | 0.0077 ± 0.0028 | 0.0089 ± 0.00053 | 0.039 ± 0.0027 |

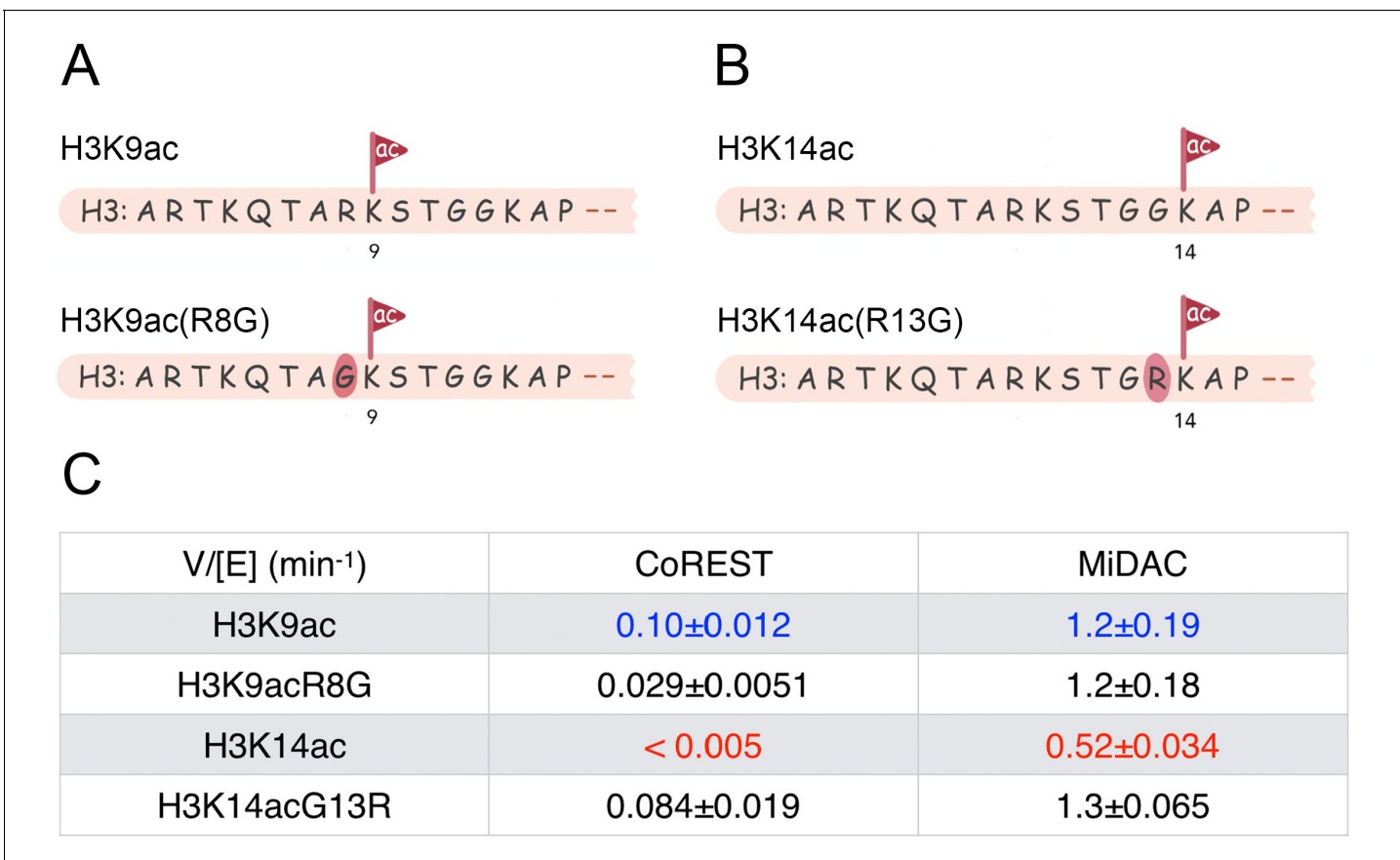

**Figure 5.** Scheme and kinetics for CoREST and MiDAC complex activity on nucleosome with mutations H3K9acR8G (n = 3 for CoREST assay, n = 5 for MiDAC assay) and 14acG13R (n = 6 for CoREST assay, n = 3 for MiDAC assay). (**A**) The sequence of H3K9acR8G comparing with WT-H3K9ac; (**B**) The sequence of H3K14acG13R comparing with WT-H3K14ac; (**C**) Table for the rate constants of different complexes on nucleosomes with H3K9ac/9acR8G/ 14ac/14acG13R. The reaction conditions were similar to the enzymatic reactions on H3K9/14/18/23/27ac nucleosomes described in *Table 1*.

The online version of this article includes the following figure supplement(s) for figure 5:

**Figure supplement 1.** MALDI-TOF spectra for H3K9acR8G H3K14acG13R 1–34 peptides, and their corresponding full length histone proteins.
**Figure supplement 2.** CoREST activity assay over nucleosomes with H3K9acR8G or H3K14acG13R.
**Figure supplement 3.** MiDAC activity assay over nucleosomes with H3K9acR8G or H3K14acG13R.

## Discussion

A principal finding of this study is that distinct HDAC1 complexes can show sharply different deacetylase activities toward different nucleosome substrates. This would appear to suggest that the unique biological consequences of the various HDAC1 complexes are not simply dependent on where they are recruited to in chromatin and by which particular transcription factor (*Watson et al., 2012b*). Rather, the different catalytic properties and site-selectivities of the HDAC1 complexes likely drive different patterns of histone deacetylation and chromatin states. Although it is possible that the closely related paralog HDAC2 replaces HDAC1 in some population of the HDAC complexes analyzed here, because of the massive overexpression of HDAC1 we believe this to be a very minor pool. Prior published structural studies of these complexes and mass spectroscopic analyses are also consistent with this idea (*Song et al., 2020*), (*Millard et al., 2016*; *Itoh et al., 2015*; *Millard et al., 2013*; *Zhang et al., 2018*), (*Portolano et al., 2014*). Furthermore, because HDAC1 and HDAC2 have very high sequence homology, we believe it is likely that HDAC1 and HDAC2 will behave in similar ways in terms of the nucleosome deacetylation assays carried out here.

The data with the CoREST complex selectivity stand out for the strong resistance to one of the five H3 tail acetylation sites, H3K14ac, investigated (*Wu et al., 2018*). It is likely not a coincidence that K14ac blocks the removal of H3K4 methylation as this could allow doubly modified H3K4me1/2 and H3K14ac tails to be immune from the silencing action of the CoREST complex. H3K4 methylation is known to be commonly found in the context of enhancer elements in DNA (*Hsu et al., 2018*), which is also associated with H3K14ac (*Karmodiya et al., 2012*), possibly insulating such K4 methylation by CoREST. Although this has not been rigorously evaluated with JMJ family demethylases, a yeast homolog JHD2 has been shown to be more weakly able to demethylate H3K4me3 in the context of H3K14ac (*Maltby et al., 2012*).

The Gly/Arg swaps at positions 8 and 13 in histone H3 clearly demonstrate that the presence of Gly13 rather than an Arg residue is a principal determinant of the meager rate of CoREST-mediated deacetylation of K14ac in nucleosome substrates. However, each of the five class I HDAC complexes examined here were relatively ineffective at deacetylating K14ac in the context of nucleosomes presumably at least partly due to Gly13. That this K14ac deselection is not true in isolated H3 indicates that the molecular recognition of Lys14ac by these HDAC complexes is altered when the H3 tail is presented to the HDAC1 active site in a chromatin context. We are unsure if this represents $k_{cat}$ or $K_M$ differences conferred by histone H3 Gly13 vs. Arg13 because of the challenges of making such measurements with nucleosome substrates. We speculate that these effects may not be reflected in overall affinities between the various HDAC complexes and particular acetylated nucleosomes, but rather are driven by subtle local differences between the HDAC complex active sites and the nucleosomal substrates. The cellular relevance of these findings is supported by the fact that H3K14ac is a relatively abundant mark at baseline (*Sidoli and Garcia, 2017*) and is rather unaffected by class I HDAC inhibitor treatment compared with H3K9ac and H3K27ac which show substantial increases with such inhibitors (*Schölz et al., 2015*). It is also possible that robust histone acetyltransferase targeting H3K14 contributes to the elevated H3K14 acetylation levels in cells.

The MiDAC complex showed far and away the speediest average deacetylation rate across the five acetyl-Lys marks looked at in nucleosomes. Impressively, MiDAC deacetylated K9ac nucleosomes (V/E 1.2 min$^{-1}$) more rapidly than K9ac in free H3 histones. Unlike CoREST, NuRD, Sin3B, and SMRT which are thought to be transcriptional silencers and recruited to chromatin by transcription factors, MiDAC is not known to be employed in this way. Furthermore, its ground state affinity to chromatin is thought to be quite high (*Itoh et al., 2015*). It is interesting that there was a modest trend for the rate of deacetylation to be highest closest to the N-terminus of H3 in the context of nucleosome substrates. We speculate that this may represent a greater accessibility of the extreme N-terminus of the H3 tail in chromatin to the HDAC1 active site in this complex.

In contrast to MiDAC, the NuRD and Sin3B complexes showed the lowest deacetylation rates with H3 acetylated nucleosomes. There are several possible reasons for this. It is plausible the NuRD and Sin3B may have a more extreme requirement for recruitment to nucleosomes by a transcription factor compared with the other HDAC complexes looked at here (*Zhang et al., 2018*), (*Walsh, 2006*). This could offer the advantage of preventing gene silencing except when triggered for by a cellular stimulus (*Millard et al., 2017*). In addition, there could be an additional macromolecular component or post-translational modification (*Wang, 2019*) lacking in our system that would

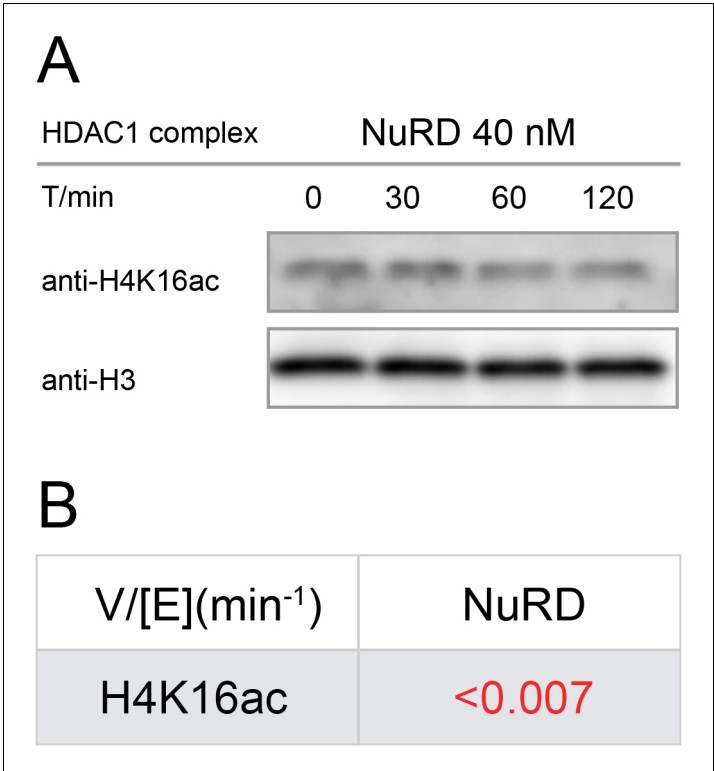

**Figure 6.** Western-blot and kinetics of NuRD complex activity on nucleosome with H4K16ac. (**A**) Western-blot based deacetylation activity assay with 40 nM NuRD complex (HDAC1-MTA1-RBBP4) on H4K16ac nucleosomes blot with site-specific acetylation antibody and anti-H3 antibody to show the overall nucleosome amount remaining unchanged (n = 3); (**B**) rate constant of NuRD complex deacetylation on H4K16ac nucleosome. The reaction conditions were similar to the enzymatic reactions on H3K9/14/18/23/27ac nucleosomes described in *Table 1*.

The online version of this article includes the following figure supplement(s) for figure 6:

**Figure supplement 1.** NuRD activity assay over H4K16ac nucleosome.

unleash the deacetylase activity or alter substrate selectivities of any of the HDAC complexes on nucleosomes (*Hudson et al., 2015*). Indeed, NuRD and the other HDAC complexes investigated here have been reported to have a variety of additional protein subunits depending on cell type and cell state (*Millard et al., 2016*), (*Lai and Wade, 2011*), (*Yang and Seto, 2008*). For the NuRD complex, however, it is established that the chromatin remodelling module is rather loosely associated, and we do not expect that its absence would greatly influence HDAC activity. In addition to roles for transcription factors and additional subunits, the NuRD or Sin3B complexes could show very high site-selectivity on one or more sites in histones beyond the acetylated-Lys that we have examined thus far in H3 or H4. Future studies will be needed to determine if one or more of these possibilities is operative.

We have focused on classical HDAC acetyl-Lys selectivities in chromatin. However, there is reason to believe that the NAD-dependent sirtuins may also show site-selectivities. Recent studies on Sirt7 have shown that this enzyme shows a strong preference for H3K36ac in the context of chromatin (*Wang et al., 2019*). It will be important in future experiments to extend the analysis of deacetylase actions across the full complement of the HDAC enzymes to create a deeper understanding of the connections between reversible acetylation in health and disease states.

# Materials and methods

## Key resources table

| Reagent type (species) or resource | Designation | Source or reference | Identifiers | Additional information |
|---|---|---|---|---|
| Gene (*X. laevis*) | Histone H2A, H2B, H4 (*X. laevis*) | DOI: 10.7554/eLife.37231 | Construct shared by Cynthia Wolberger (Johns Hopkins) | |
| Gene (*X. laevis*) | Histone gH3 (*X. laevis*) | DOI: 10.7554/eLife.37231 DOI: 10.1021/ja205630g | Construct shared by Dirk Schwarzer (Tübingen) | |
| Gene (*S. aureus*) | F40-sortase (Originally mutated from *S. aureus* sortase A) | DOI: 10.7554/eLife.37231 DOI: 10.1021/ja205630g | Construct shared by Dirk Schwarzer (Tübingen) | |
| Gene (*E. coli*) | 146 bp 601 DNA (Widom sequence for *E. coli* expression) | DOI: 10.7554/eLife.37231 | Construct shared by Song Tan (Penn State) | |
| Strain, strain background (*E. coli*) | BL21 Rosetta (DE3) pLysS (*E. coli*) | Sigma-Aldrich | Cat# 70956–4 | Competent cells |
| Cell line (*Homo-sapiens*) | HEK293F (*Homo-sapiens*) | Thermo Fisher Scientific | RRID:CVCL_6642 | Transfected with plasmids to express different HDAC1 complexes. Mycoplasma tested negative. Authenticated by the venfor. Not on ICLAC misidentified cell line registry. |
| Antibody | Anti-H3K9ac (Rabbit) | Cell Signaling | Cat# 9671, RRID:AB_331532 | WB (1:2000) |
| Antibody | Anti-H3K14ac (Rabbit) | EMP Millipore | Cat# 07–353, RRID:AB_310545 | WB (1:2000) |
| Antibody | Anti-H3K18ac (Rabbit) | EMP Millipore | Cat# 07–354, RRID:AB_ 441945 | WB (1:2000) |
| Antibody | Anti-H3K23ac (Rabbit) | EMP Millipore | Cat# 07–355, RRID:AB_310546 | WB (1:2000) |
| Antibody | Anti-H3K27ac (Rabbit) | Cell Signaling | Cat# 8173, RRID:AB_10949503 | WB (1:2000) |
| Antibody | Anti-H3 (Rabbit) | Abcam | Cat# ab1791, RRID:AB_302613 | WB (1:5000) |
| Antibody | Anti-H4K16ac (Rabbit) | Active Motif | Cat# 39167, RRID:AB_2636968 | WB (1:5000) |
| Antibody | HRP conjugated anti-Rabbit secondary (Goat) | Cell Signaling | Technology Cat# 7074, RRID:AB_2099233 | WB (1:2000) |
| Peptide, recombinant protein | HDAC1/HSP70 | BPS Bioscience | Cat# 50051 | |
| Peptide, recombinant protein | Nucleosome with H4K16ac | Epicypher | Cat# 16–0354 | |
| Chemical compound | 20 Fmoc-amino acids | EMP Millipore DOI: 10.7554/eLife.37231 | Fmoc-Gly-OH as example: Cat# 852001 | |
| Software, algorithm | ImageJ | DOI: 10.7554/eLife.37231 Download from imagej.nih.gov/ij/ | RRID:SCR_003070 | |
| Software, algorithm | GraphPad Prism Eight | DOI: 10.7554/eLife.37231 Download from www.graphpad.com | RRID:SCR_002798 | |

## Reagents

20 amino acids were purchased from EMP Millipore. Site specific primary antibodies include anti-H3K9ac (Cell Signaling), anti-H3K14ac (EMP Millipore), anti-H3K18ac (EMP Millipore), anti-H3K23ac (EMP Millipore), anti-H3K27ac (Cell Signaling), anti-H4K16ac (Active Motif) antibodies, as well as anti-histone H3 (Abcam). Secondary antibody as HRP-conjugated anti-rabbit antibody from Cell Signaling. H4K16ac nucleosome was purchased from Epicypher. The free enzyme HDAC1 (in the complex with HSP70) is purchased from BPS Bioscience.

## The synthesis of depsipeptide Fmoc-Thr(OtBu)-glycolic acid (TOG)

The depsipeptide Fmoc-Thr(OtBu)-glycolic acid was synthesized based on a reported two-step protocol (*Williamson et al., 2012*).

## Peptide synthesis

All the histone H3 peptides 1–34 containing site-specific lysine acetylation (Kac) were synthesized by Fmoc-SPPS by Fmoc-Gly-Wang resin. For 0.2 mmol resin, 0.8 mmol (four eq) Fmoc-amino acid, 3.2 mmol (3.75 eq) HATU and 1.6 mmol (eight eq) N-methylmorpholine (NMM) in 8 mL DMF was added to do the peptide coupling for 90 min, followed by the deprotection of Fmoc with 20% piperidine in dimethylformamide (DMF) for 15 min. The resin was then completely washed with DMF before entering the next coupling cycle. The first depsipeptide TOG was coupled following a similar protocol. After all the coupling steps and deprotection steps, the resin was washed by DMF and dichloromethane (DCM) to dry. Reagent B (5% water, 5% phenol, 2.5% triisopropyl silane (TIPS), 87.5% trifluoroacetic acid (TFA)) for 2.5 hr at room temperature. The crude peptide was concentrated before the addition of cold diethyl ether to precipitate and dried over nitrogen gas flow. After being dissolved in 10% acetonitrile ($CH_3CN$)-water, the crude peptides were further purified by reversed phase HPLC with the C18 column (Varian Dynamax Microsorb 100), with a gradient of 5% $CH_3CN$/0.05% TFA in $H_2O$/0.05% TFA for 2 min and a linear gradient from 5% $CH_3CN$/0.05% TFA to 32% $CH_3CN$/0.05% TFA over 25 min, and the flow rate is 10 mL/min. All the synthetic peptides were lyophilized to powders, followed by the characterization by MALDI-TOF at the Molecular Biology Core Facilities of Dana Farber Cancer Institute (*Figure 2—figure supplement 1*, *Figure 5—figure supplement 1*).

> H3K9ac(1-34) TOG [M + H]+ calculated as m/z 3494.9, observe at m/z 3493.0;
> Sequence: ARTKQTARKS-TGGKAPRKQL-ATKAARKSAP-A-TOG-G
> H3K14ac(1-34) TOG [M + H]+ calculated as m/z 3494.9, observed at m/z 3493.5;
> H3K18ac(1-34) TOG [M + H]+ calculated as m/z 3494.9, observed at m/z 3492.9;
> H3K23ac(1-34) TOG [M + H]+ calculated as m/z 3494.9, observed at m/z 3493.7;
> H3K27ac(1-34) TOG [M + H]+ calculated as m/z 3494.9, observed at m/z 3493.1;
> H3K9acR8G(1-34) TOG [M + H]+ calculated as m/z 3394.8, observed at m/z 3393.8;
> H3K14acG13R(1-34) TOG [M + H]+ calculated as m/z 3594.1, observed at m/z 3591.7;

## Expression and purification of F40 sortase from *E. coli*

F40 sortase was purified in a manner reported previously (*Wu et al., 2018*). The pET21-F40-sortase plasmid was used for the transformation of the *E. coli* Rosetta (DE3) pLysS strain. A single colony from the LB agar plate was inoculated into 5 mL LB media with 100 mg/L ampicillin at 37 °C overnight, and subsequently inoculated into 1 L large 2YT culture at 37 °C until $OD_{600}$ reached 0.6. F40 sortase was then induced with 0.25 mM IPTG for 4 hr at 37 °C. The cells were harvested by centrifugation at 4,000 rpm for 15 min and the cell-pellet was resuspended in Lysis buffer containing 20 mM Tris, 0.1% Triton X-100 and 1 mM phenylmethylsulfonyl fluoride (PMSF) at pH 7.5. The cells were then lysed by passing through a french press cell disrupter at about 1,000 psi three times, followed by centrifuging the lysate at 20,000 g for 40 min. The supernatant was loaded to a Ni-NTA Sepharose 6 Fast Flow (GE Healthcare) resin, followed by repeated wash with Wash buffer containing 20 mM Tris, 500 mM NaCl, 20 mM imidazole and 1 mM PMSF at pH 7.5. The F40 sortase was then eluted with Elution buffer containing 20 mM Tris, 500 mM NaCl, 400 mM imidazole and 1 mM PMSF at pH 7.5 in different fractions. All the fractions were analyzed by SDS-PAGE gel and fractions containing F40 sortase were pooled together and dialyzed against Dialysis buffer with 20 mM Tris, 150 mM NaCl, and 5 mM $CaCl_2$ at pH 7.5. After dialysis, F40 sortase (>95% pure based on coomassie

stained SDS-PAGE) was concentrated using an Amicon Ultra spin column (10K MWCO, EMD Millipore) to about 1 mM final concentration, and then aliquoted and stored at −80 ˚C for future usage.

## Expression of WT-histone H2A, H2B, H4, and histone core region gH3

Bacterial expression and purification of *X. laevis* core histones H2A, H2B, globular H3 (gH3; amino acids 34–135 with the first Met cleaved by *E. coli*), and H4 were performed using a previously reported method (*Piotukh et al., 2011*), (*Luger et al., 1997*). Please note that we used gH3 containing Cys110 for K23ac and K27ac semi-synthesis and Ala110 for K9ac, K14ac, K18ac semi-synthesis (*Shimko et al., 2011*) and the impact of this aa110 difference in the nucleosomes derived from these histone H3 isoforms is assumed to be negligible (*Wilkinson and Gozani, 2014*). Consequently, H3 C110A has been widely employed in nucleosome chemical biology (*Jani et al., 2019*) and structural studies (*Gatchalian et al., 2017*). The pET23-gH3 plasmid was used to transform *E. coli* Rosetta (DE3) pLysS strain, and a single colony was inoculated into 1 L LB media with 100 mg/L ampicillin at 37 ˚C. The culture was let to shake and grow at 37 ˚C until $OD_{600}$ reached 0.6. Then, gH3 was induced with 0.5 mM IPTGat 37 ˚C for 3 hr. The cells were then harvested by centrifugation at 4,000 rpm for 15 min and the cell pellet was resuspended in Histone Wash Buffer (50 mM Tris-HCl at pH 7.5, 100 mM NaCl, 1 mM EDTA, 5 mM 2-mercaptoethanol (BME), 0.2 mM phenylmethylsulfonyl fluoride (PMSF) with 1% Triton X-100). The cells were then lysed by passing through a french press cell disrupter at about 1,000 psi three times, followed by centrifugation of the cell lysate at 20,000 g for 40 min. After discarding the supernatants, the pellets were washed with Histone Wash Buffer with 1% Triton X-100 once and Histone Wash Buffer without Triton X-100 twice. The pellets were dissolved in Histone Solubilization Buffer (7 M guanidinium hydrochloride, 20 mM Tris at pH 7.5 and 10 mM dithiothreitol). After centrifuging at 20,000 g for 15 min, the supernatant was dialyzed three times against IEX (ion exchange) buffer (7 M urea, 10 mM Tris pH 7.8, 1 mM EDTA, 0.2 mM PMSF and 5 mM BME). The resulting gH3 solution in IEX buffer was diluted 5-fold in IEX buffer before being loaded onto a tandem Q-SP column (GE healthcare, HiTrap Q HP and HiTrap SP HP, 5 mL each). Both Q and SP column were first equilibrated and washed with IEX buffer containing 100 mM NaCl. Then, gH3 was eluted from the 100 mM - 500 mM NaCl with 100 mM increment. The fractions containing gH3 (>90% pure by coomassie stained SDS-PAGE) were pooled after analyzing each fraction by SDS-PAGE. The combined IEX solution was then dialyzed three times against pure water containing 2 mM BME (3.5K MWCO, Spectra/Por) for 6 hr each. The solution was finally concentrated with an Amicon Ultra spin column (3.5 K MWCO, EMD Millipore) to 10 mL (~100 μM) and stored at 4 ˚C until usage.

## F40 sortase catalyzed histone H3 semi-synthesis

The histone H3 N-terminal peptide (aa1-34 depsipeptide, 1 mM) and gH3 (aa34-135,~70 μM) was mixed in Sortase Reaction Buffer (50 mM HEPES at pH 7.5, 150 mM NaCl and 5 mM $CaCl_2$). The reaction was initiated by the addition of F40 Sortase (300 μM) and was incubated at 37 ˚C for overnight. The reaction mixture was centrifuged to precipitate H3 with some sortase, which was then dissolved in 10 mL IEX buffer (7 M urea, 10 mM Tris at pH 7.8, 1 mM EDTA and 5 mM BME). The solution was loaded onto a SP column (GE healthcare, HiTrap SP HP, 1 mL) preequilibrated with IEX buffer containing 100 mM NaCl. After washing with IEX buffer containing 100 mM NaCl, H3 was eluted with a gradient of IEX buffer containing increasing NaCl from 100 to 500 mM. All the fractions were analyzed by SDS-PAGE, and the fractions with pure ligated H3 product (>90% based on coomassie stained SDS-PAGE gels) were pooled and dialyzed three times against pure water with 2 mM BME (3.5K MWCO, Spectra/Por). The solution was then concentrated using an Amicon Ultra spin column (10 K MWCO, EMD Millipore), and was lyophilized to a white powder for storage at −80 ˚C until usage. Semi-synthesized histone H3 structures were characterized by MALDI-TOF Mass Spectrometry ((matrixed with CHCA) at the Molecular Biology Core Facilities of Dana Farber Cancer Institute (*Figure 2—figure supplement 2*, *Figure 5—figure supplement 1*).

H3K9ac(C110A): $[M + H]^+$ calculated as m/z 15280.8, observed as m/z 15274.3;
H3K14ac(C110A): $[M + H]^+$ calculated as m/z 15280.8, observed as m/z 15278.9;
H3K18ac(C110A): $[M + H]^+$ calculated as m/z 15280.8, observed as m/z 15271.6;
H3K23ac: $[M + H]^+$ calculated as m/z 15312.8, observed as m/z 15309.6;
H3K27ac: $[M + H]^+$ calculated as m/z 15312.8, observed as m/z 15308.6;

H3K9acR8G(C110A): $[M + H]^+$ calculated as m/z 15181.6, observed as m/z 15175.1;
H3K14acG13R(C110A): $[M + H]^+$ calculated as m/z 15379.9, observed as m/z 15377.1.

## Octamer refolding and nucleosome reconstitution

The octamer refolding and nucleosome assembly were performed following a previous report (*Dyer et al., 2004*). All four histone proteins H2A, H2B, H3 and H4, each two copies, were dissolved in Unfolding Buffer (7 M guanidine, 20 mM Tris-HCl at pH 7.5 and 10 mM DTT) and quantified by reading $A_{280}$. The mixed solution with a molar ratio at 1.1: 1.1: 1:1 one was then dialyzed three times against high salt buffer (20 mM Tris at pH 7.5, 2.0 M NaCl, 1 mM EDTA and 5 mM BME). The octamer was purified by size exclusion chromatography using a Superdex 200 10/300 GL column (GE Healthcare). In addition, 146 Widom 601 DNA was prepared by previously reported methods used for nucleosome reassembly (*Luger et al., 1997*). The 146 bp DNA was obtained from the restriction digests of a specially designed DNA plasmid from *E. coli* and purified by precipitation with polyethylene glycol. The histone octamer (*Figure 2—figure supplement 3B*, using H3K9ac and H3K14ac as examples) and Widom DNA were mixed at a 1:1 molar ratio at high salt buffer (10 mM Tris at pH 7.5, 2.0 M KCl, 1 mM EDTA and 1 mM DTT) with a final concentration of 7 μM DNA. Continuous-flow gradient dialysis was used to lower the salt concentration from the mixture to low salt buffer (10 mM Tris at pH 7.5, 250 mM KCl, 1 mM EDTA and 1 mM DTT) by a Minipuls two peristaltic pump (Gilson). The final mixture was subjected to HPLC (Waters, 1525 binary pump, 2489 UV-Vis detector). HPLC using a TEKgel DEAE ion exchange column was employed to purify the final nucleosome product. The liquid phase buffers A and B were as follow: A. TES 250 (10 mM Tris 7.5, 250 mM KCl, 0.5 mM EDTA), B. TES 600 (10 mM Tris 7.5, 600 mM KCl, 0.5 mM EDTA), were used to generate with a 0% B wash for 12 min and a linear gradient from 25% to 75% B over 30 min at 1 mL/min flow rate. The fractions containing the nucleosome were pooled together and dialyzed against a TCS Buffer (20 mM Tris at pH 7.5 and 1 mM DTT, no EDTA) and concentrated to about 20 μM using an Amicon Ultra spin column (10K MWCO, EMD Millipore). The final nucleosome products were analyzed by native TBE gels and all showed over 95% purity based on the fact that free DNA band fluoresces 2.5-fold more brightly than the nucleosome DNA band in this setting (*Shahian and Narlikar, 2012*).

## Expression and purification of HDAC complexes

The expression and purification of HDAC complexes were performed as described previously (*Song et al., 2020*), (*Millard et al., 2016*), (*Itoh et al., 2015*), (*Watson et al., 2012b*), (*Portolano et al., 2014*). For each complex, three plasmids corresponding to the different components of the HDAC complex were co-transfected into HEK293F cells: LSD1 (1–852), HDAC1 (1–482), and Flag tagged CoREST1 (86–485) for the CoREST complex; Flag tagged MTA1 (1–715), HDAC1 (1–482), and RBBP4 (1–425) for the NuRD complex; Flag tagged Sin3B (1–1162), HDAC1 (1–482), RBBP4 (1–425) for the Sin3B complex; Flag tagged MIDEAS (628-887), HDAC1 (1–482), DNTTP1 (1–329) for the MiDAC complex; Flag tagged GPS2-NCoR2 chimera ((1-49) - (220-480)), HDAC3 (1–428), and TBL1 (1–526) for the SMRT complex. After cell lysis by sonication, the complexes were purified by FLAG tag immunoaffinity chromatography and liberated from the column with TEV protease. These mixtures were further purified by gel filtration on either a Superdex 200 Increase 10/300 GL column (MiDAC, SMRT) or a Superose 6 10/300 GL column (CoREST, NuRD, Sin3B). The purified complexes were analyzed by SDS-PAGE stained with coomassie and concentrated to 3–5 μM (*Figure 3—figure supplement 2B*). After the addition of glycerol to 10%, the complexes are stored at −80°C until further use (*Wu et al., 2018*). The final HDAC complexes were analyzed by SDS-PAGE gel stained with coomassie and each showed greater than 60% purity.

## Analysis of deacetylation of acetylated nucleosomes and acetylated histone H3s

For all the deacetylation assays, the semi-synthetic H3K9/14/18/23/27ac histone H3 proteins (1.0 μM) or the corresponding nucleosomes (100 nM) assembled in vitro were incubated with different HDAC1 and HDAC3 complexes at different concentrations, under a reaction buffer containing 50 mM HEPES at pH 7.5, 100 mM KCl, 100 μM IP6 (*Watson et al., 2016*), and 0.2 mg/mL BSA at 37 °C. The complexes include CoREST, NuRD, Sin3B, MiDAC, and SMRT, together with HDAC1 enzyme

not involved in specific complex binding from commercial source (BPS Bioscience). The reaction time was counted, and multiple samples were taken at different time points. Typically, 10 µL aliquots of the reaction were quenched with gel loading buffer made by mixing 2 µL 80 mM EDTA and 4 µL 4 x Laemmli sample buffer at each time point. The samples were then boiled for 3 min at 95 ˚C and loaded onto a 15% SDS-PAGE gel (160 Volts for 35 min). After transferring to nitrocellulose membrane, site-specific antibodies for acetylated H3, such as for anti-H3K9ac, anti-H3K14ac, anti-H3K18ac, anti-H3K23ac and anti-H3K27ac, as well as total H3 antibody, anti-H3, were used to blot the membrane. The specificities of the primary antibodies, anti-H3K23ac and anti-H3K27ac, were validated using histone H3K9ac as a negative control (*Figure 3—figure supplement 1*), while the specificities of the primary antibodies anti-H3K9ac, anti-H3K14ac and anti-H3K18ac have been confirmed previously (*Wu et al., 2018*). After treating with the ECL substrate (Bio-Rad), the blotted membrane was and visualized by the G:BOX mini gel imager (Syngene), and then quantified using ImageJ. The intensity values from ImageJ were fit to a single-phase exponential decay curve software (GraphPad Prism Eight). Then, the kinetic data V/E were calculated for nucleosome assays, while normalized V/E were calculated for histone assays (Two H3 molecules to one nucleosome ratio was taken into account for calculation).

## Analysis of deacetylation of H4K16ac nucleosomes

For the H4 deacetylation assays, the commercial H4K16ac nucleosomes (100 nM) was incubated with 40 nM NuRD complexes in a reaction buffer containing 50 mM HEPES at pH 7.5, 100 mM KCl, 100 µM IP6, and 0.2 mg/mL BSA at 37 ˚C. Multiple samples were taken at different time points and analyzed by Western Blot with anti-H4K16ac antibody, following similar steps of the deacetylation assays of the H3Kac nucleosomes.

## Acknowledgements

The authors would like to thank the PROTEX cloning service, University of Leicester, for producing the constructs used to make HDAC complexes. The authors thank James Lee (Molecular Biology Core Facilities of Dana Farber Cancer Institute) for the assistance of MALDI, Yi Zheng for illustrations, Ben Martin and Sam Whedon for advice, and Taylor Kay for technical support. The authors acknowledge NIH, LLS, and the Wellcome Trust for financial support.

## Additional information

### Competing interests

Philip A Cole: Senior editor, *eLife*. The other authors declare that no competing interests exist.

### Funding

| Funder | Grant reference number | Author |
|---|---|---|
| NIH | GM62437 | Philip A Cole |
| Leukemia and Lymphoma Society | SCOR | Philip A Cole |
| Wellcome Trust | 100237/Z/12/Z | John W R Schwabe |

The funders had no role in study design, data collection and interpretation, or the decision to submit the work for publication.

### Author contributions

Zhipeng A Wang, Conceptualization, Data curation, Investigation, Methodology, Writing - original draft, Writing - review and editing; Christopher J Millard, Chia-Liang Lin, Conceptualization, Resources, Investigation, Methodology, Writing - review and editing; Jennifer E Gurnett, Conceptualization, Resources, Investigation, Writing - review and editing; Mingxuan Wu, Conceptualization, Formal analysis, Investigation, Methodology, Writing - review and editing; Kwangwoon Lee, Investigation, Writing - review and editing; Louise Fairall, Conceptualization, Resources, Supervision, Project

administration, Writing - review and editing; John WR Schwabe, Conceptualization, Formal analysis, Supervision, Funding acquisition, Methodology, Project administration, Writing - review and editing; Philip A Cole, Conceptualization, Formal analysis, Supervision, Funding acquisition, Writing - original draft, Project administration, Writing - review and editing

### Author ORCIDs
Zhipeng A Wang ⓘ https://orcid.org/0000-0002-5693-7359
Kwangwoon Lee ⓘ https://orcid.org/0000-0002-2021-5186
John WR Schwabe ⓘ https://orcid.org/0000-0003-2865-4383
Philip A Cole ⓘ https://orcid.org/0000-0001-6873-7824

### Decision letter and Author response
Decision letter https://doi.org/10.7554/eLife.57663.sa1
Author response https://doi.org/10.7554/eLife.57663.sa2

## Additional files

### Supplementary files
• Transparent reporting form

### Data availability
Data has been uploaded to Dryad under the DOI: https://doi.org/10.5061/dryad.x0k6djhgc.

The following dataset was generated:

| Author(s) | Year | Dataset title | Dataset URL | Database and Identifier |
|---|---|---|---|---|
| Wang ZA, Millard CJ, Lin C-L, Gurnett JE, Wu M, Lee K, Fairall L, Schwabe JW, Cole PA | 2020 | Diverse nucleosome site-selectivity among histone deacetylase complexes | https://doi.org/10.5061/dryad.x0k6djhgc | Dryad Digital Repository, 10.5061/dryad.x0k6djhgc |

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
