## [Decision Letter]

Thank you for submitting your article "Diverse nucleosome site-selectivity among histone deacetylase complexes" for consideration by *eLife*. Your article has been reviewed by two peer reviewers, and the evaluation has been overseen by Wilfred van der Donk as Reviewing Editor and John Kuriyan as the Senior Editor. The reviewers have opted to remain anonymous.

The reviewers have discussed the reviews with one another and the Reviewing Editor has drafted this decision to help you prepare a revised submission.

Summary:

This work is an extension of previous work from the Cole lab that reported (1) the use of F40 sortase to generate semi-synthetic acetylated H3, (2) its incorporation in nucleosomes, and (3) assays with the LSD1-CoREST-HDAC1 complex. In this work, the authors expand upon this prior work by testing various acetylated nucleosomes against 4 different HDAC1-containing and one HDAC3-containing protein complex (CoREST, NuRD, Sin3B, MiDAC and SMRT). Interestingly, the authors observe that H3K14ac is a poor substrate for the CoREST and Sin3B complexes, and speculate that this may be due to the presence of Gly13 at the P2 position. They demonstrate that mutating Gly13 to Arg13 leads to faster deacetylation by the CoREST complex. Another interesting discovery is that HDAC1 activity at various H3 lysines is modulated by its specific complex. This work is important towards obtaining a molecular level understanding of how various HDAC complexes act on chromatin depending on both enzyme context and substrate context. The reviewers feel this Research Advance is suitable for publication in *eLife* after minor clarifications.

Revisions:

1) Enzyme activity that is modulated by its context in various complexes is perhaps not too surprising, but gaining information of such modulation is important and valuable. In the current system, the complexes likely bind nucleosomes at different sites and/or allosterically regulate k_cat_ or K_M_. One aspect of this study that is not well-explained is why complexes were chosen to be limited to three proteins. NuRD, for example, has six known components. The authors should address the choice for a limited set of proteins, and why three components were chosen in each case.

2) The authors propose that HDAC1 activity is inhibited by G13 in the CoREST complex. However, it is unclear why the same residue adjacent to K14ac in the HDAC1 active site would have vastly different effects (CoREST vs MIDAC1). Do the authors have any insights into whether Gly13Arg induces activity due to an effect on k_cat_ or K_M_ (or both)? In future work, determination of binding constants would be valuable.

3) Many of the semisynthetic H3 proteins have additional large unlabeled peaks in the MALDI-TOF mass spectra. These are less visible in the charge states in the ESI spectra due to the large mass range shown but also appear to show up in the western blots (Figure 3—figure supplement 1). The authors should clearly label the additional peaks. It also appears there would be free DNA in the nucleosome preparations. Free DNA is known to inhibit enzymatic activities in some instances, for e.g. for p300. It would be valuable to add a statement to the manuscript as to the percent purity of the nucleosome substrates.

4) Similarly the protein complexes in Figure 3—figure supplement 2B are not homogenous. While purifying complexes using epitope tags from eukaryotic cells is unlikely to yield perfectly homogenous preps, it is important for the authors to mention the percent purity of each complex. Given point 1 above, can they comment whether other components are entirely missing from these complexes and do not have an effect on activity?

5) One concern is that HDAC1 is often exchanged with HDAC2 in many complexes, including NuRD. It appears that the authors did not interrogate HDAC2 directly, but can they confirm that no residual HDAC2 was present in their complexes? Varying degrees of HDAC1/2 may contribute toward some of the differences seen in kinetics data.

6) The authors extend their discovery to the discussion that H3K14ac and H3K4me2 may co-exist at enhancers to prevent CoREST activity. This may be the case, however other enzymes such as KDM4 (Jumonji-domain containing lysine specific demethylase 4) are equally able to demethylate H3K4me2. There is no evidence that these KDMs are sensitive to inhibition by H3K14ac, or that they are excluded from enhancers while CoREST is recruited to enhancers. Therefore, the presented argument may need some additional discussion. Could the presence of H3K14ac at high basal levels equally likely reflect higher degrees of basal HAT activity at this site?

---

## [Author Response]

Revisions:1) Enzyme activity that is modulated by its context in various complexes is perhaps not too surprising, but gaining information of such modulation is important and valuable. In the current system, the complexes likely bind nucleosomes at different sites and/or allosterically regulate k_cat_ or K_M_. One aspect of this study that is not well-explained is why complexes were chosen to be limited to three proteins. NuRD, for example, has six known components. The authors should address the choice for a limited set of proteins, and why three components were chosen in each case.

There is no doubt that there are multiple versions of each of the endogenous HDAC1/HDAC2 complexes. These differences may be developmentally regulated, cell cycle dependent, tissue specific etc. As the reviewers point out, in general, 3 proteins were included for the complexes studied here, although for the SMRT complex, one of the proteins is a chimeric GPS2-NCOR2 fusion. The selection of what we regard as the core complexes comes from almost a decade of experience of a combination of biochemical and structural studies to develop stable, enzymatically functional complexes that can be purified and can interact efficiently with nucleosomes. The many details of how these complexes have been arrived at and are produced are described in a series of Schwabe lab publications cited in the manuscript that reflect the fruits of these laborious efforts to generate them. In general, the two non-HDAC proteins in each case were selected based on the following criteria: 1) A set of proteins that included both well-established HDAC and nucleosome binding partners for a given complex, 2) Efficient transient co-expression of soluble proteins in HEK293F that could stay associated by immunoaffinity chromatography and lead to relatively pure and concentrated complexes in peak fractions (>50% purity), 3) The ability of the core complexes to be reproducibly isolated as monodisperse peaks in appropriate stoichiometries after size exclusion chromatography. We recognize that it is certainly possible that other subunits, not included in these core complexes, could alter catalytic efficiencies and/or substrate selectivities. We included this point for NuRD in the original manuscript but now make the point more broadly in the revised manuscript. For the NuRD complex, it is established that the chromatin remodelling module is rather loosely associated, and we do not expect that its absence would greatly influence HDAC activity. Moreover, NuRD complexes in which MTA1 is substituted by PWWP2A/B do not associate with the chromatin remodellers. We have revised the manuscript to reflect these additional points.

2) The authors propose that HDAC1 activity is inhibited by G13 in the CoREST complex. However, it is unclear why the same residue adjacent to K14ac in the HDAC1 active site would have vastly different effects (CoREST vs MIDAC1). Do the authors have any insights into whether Gly13Arg induces activity due to an effect on k_cat_ or K_M_ (or both)? In future work, determination of binding constants would be valuable.

We have not measured k_cat_ and K_M_ values due to the challenges involved of attempting such measurements in this complex system including the requirement of working with high concentrations of nucleosomes and our reliance on western blotting and disappearance of acetyl-Lys signals. We do think measuring binding affinities is feasible and plan to do this with fluorescently labeled reagents in future work. Our speculation is that the nucleosome binding affinity will not be strongly affected by H3 Gly13 and that differences in CoREST vs. MiDAC complex deacetylation rates will be related to precise active site complementarity between these catalysts. We have added these comments to the manuscript.

3) Many of the semisynthetic H3 proteins have additional large unlabeled peaks in the MALDI-TOF mass spectra. These are less visible in the charge states in the ESI spectra due to the large mass range shown but also appear to show up in the western blots (Figure 3—figure supplement 1). The authors should clearly label the additional peaks. It also appears there would be free DNA in the nucleosome preparations. Free DNA is known to inhibit enzymatic activities in some instances, for e.g. for p300. It would be valuable to add a statement to the manuscript as to the percent purity of the nucleosome substrates.

We agree that there are some minor impurities in some of the semisynthetic histone H3 MALDI mass spectra (we have not performed ESI in this case), particular in the molecular weight range of the globular H3 fragment. We have now labeled these with asterisks in the revised figures. We think it is unlikely that these impurities at the histone stage will interfere with the nucleosome deacetylation assays as they are likely to be removed at the octamer and/or nucleosome purification stages. We also emphasize that the relative similarity of the deacetylation rates with each of these free acetyl-histone H3 substrates indicate that the impurities probably do not have a large impact on the general activity of HDAC complexes. Regarding the purities of the mononucleosomes with regard to free DNA, we agree that the presence of free DNA could potentially interfere with the deacetylase assays. However, these native DNA gels stained with ethidium bromide indicate <5% free DNA in molar terms. The free DNA band visualized is 10% or less of the relative fluorescence intensity for the nucleosomes among the different preparations. But such analysis overestimates the impurity. The more valid calculation of purity takes into account that free DNA vs. nucleosome bound DNA fluoresces about 2.5 times more brightly due to the packaging with the histone proteins under the conditions of these gels (Shahian and Narlikar, Methods in Mol. Biol. 2012). Thus, we believe that the contaminating free DNA in these preparations is acceptably low for these enzymatic assays. We have summarized each of these points in the revised manuscript.

4) Similarly the protein complexes in Figure 3—figure supplement 2B are not homogenous. While purifying complexes using epitope tags from eukaryotic cells is unlikely to yield perfectly homogenous preps, it is important for the authors to mention the percent purity of each complex. Given point 1 above, can they comment whether other components are entirely missing from these complexes and do not have an effect on activity?

We agree that these HDAC complexes are not completely pure. We have now estimated the purities of each complex based on coomassie staining of the impurity bands relative to the desired specific protein bands within the complex. These estimated purities are as flows: CoREST (84%), NuRD (93%), Sin3B (63%), MIDAC (84%), and SMRT (99%). It is possible that impurities seen or unseen could affect the activities of these complexes but we do not have a simple way to analyze this. However, for various nucleosome deacetylation assays, we have used more than one independent HDAC complex preparation for enzymatic measurements and have seen consistent deacetylase results across different preps.

5) One concern is that HDAC1 is often exchanged with HDAC2 in many complexes, including NuRD. It appears that the authors did not interrogate HDAC2 directly, but can they confirm that no residual HDAC2 was present in their complexes? Varying degrees of HDAC1/2 may contribute toward some of the differences seen in kinetics data.

As the reviewers note, HDAC1 and HDAC2 show overall about 80% amino acid sequence identity and can be interchangeably incorporated into multiprotein complexes under endogenous conditions. Because the transient transfections of HEK293F cells with HDAC1 plasmid leads to massive overexpression of HDAC1 relative to endogenous levels of HDAC1 or HDAC2, we think it is highly likely that the vast majority of HDAC isoform in these complexes is HDAC1. Prior published structural studies of these complexes from the Schwabe lab and mass spectroscopic analyses of tryptic digests are also consistent with this idea. Having said that, because of their very high sequence homology, we believe it is likely that HDAC1 and HDAC2 will behave in similar ways in terms of the nucleosome deacetylation assays carried out here. We have summarized these HDAC2 related points in the revised manuscript.

6) The authors extend their discovery to the discussion that H3K14ac and H3K4me2 may co-exist at enhancers to prevent CoREST activity. This may be the case, however other enzymes such as KDM4 (Jumonji-domain containing lysine specific demethylase 4) are equally able to demethylate H3K4me2. There is no evidence that these KDMs are sensitive to inhibition by H3K14ac, or that they are excluded from enhancers while CoREST is recruited to enhancers. Therefore, the presented argument may need some additional discussion. Could the presence of H3K14ac at high basal levels equally likely reflect higher degrees of basal HAT activity at this site?

It is certainly possible that high HAT activity could also drive elevated H3K14ac as suggested by the reviewers. We believe that the reviewers may be referring to KDM5 rather than KDM4 as an JMJ family H3K4me demethylase (KDM4 is better known at targeting other H3 sites). Interestingly, there is a report that the JMJ H3K4me demethylase paralog in yeast appears to be more weakly able to demethylate in the context of H3K14ac (Maltby et al., 2012). We have emphasized these points more clearly in the revised manuscript.